# Mapping Images to Scene Graphs with Permutation-Invariant Structured Prediction

**Roei Herzig**\*
Tel Aviv University
roeiherzig@mail.tau.ac.il

**Moshiko Raboh**\*
Tel Aviv University
mosheraboh@mail.tau.ac.il

**Gal Chechik**
Bar-Ilan University, NVIDIA Research
gal.chechik@biu.ac.il

**Jonathan Berant**
Tel Aviv University, AI2
joberant@cs.tau.ac.il

**Amir Globerson**
Tel Aviv University
gamir@post.tau.ac.il

## Abstract

Machine understanding of complex images is a key goal of artificial intelligence. One challenge underlying this task is that visual scenes contain multiple inter-related objects, and that global context plays an important role in interpreting the scene. A natural modeling framework for capturing such effects is structured prediction, which optimizes over complex labels, while modeling within-label interactions. However, it is unclear what principles should guide the design of a structured prediction model that utilizes the power of deep learning components. Here we propose a design principle for such architectures that follows from a natural requirement of permutation invariance. We prove a necessary and sufficient characterization for architectures that follow this invariance, and discuss its implication on model design. Finally, we show that the resulting model achieves new state-of-the-art results on the *Visual Genome* scene-graph labeling benchmark, outperforming all recent approaches.

## 1 Introduction

Understanding the semantics of a complex visual scene is a fundamental problem in machine perception. It often requires recognizing multiple objects in a scene, together with their spatial and functional relations. The set of objects and relations is sometimes represented as a graph, connecting objects (nodes) with their relations (edges) and is known as a *scene graph* (Figure 1). Scene graphs provide a compact representation of the semantics of an image, and can be useful for semantic-level interpretation and reasoning about a visual scene [11]. Scene-graph prediction is the problem of inferring the joint set of objects and their relations in a visual scene.

Since objects and relations are inter-dependent (e.g., a person and chair are more likely to be in relation "sitting on" than "eating"), a scene graph predictor should capture this dependence in order to improve prediction accuracy. This goal is a special case of a more general problem, namely, inferring multiple inter-dependent labels, which is the research focus of the field of structured prediction. Structured prediction has attracted considerable attention because it applies to many learning problems and poses

---

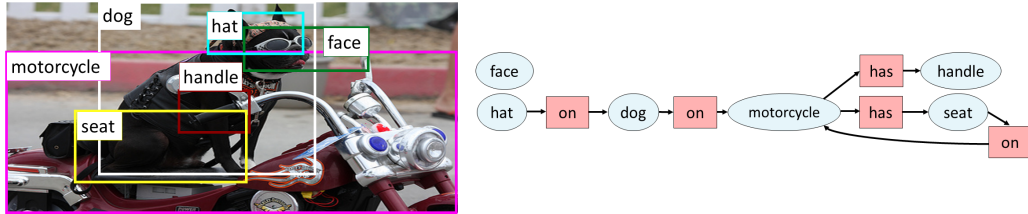

Figure 1: An image and its scene graph from the Visual Genome dataset [15]. The scene graph captures the entities in the image (nodes, blue circles) like *dog* and their relations (edges, red circles) like ⟨hat, on, dog⟩.

unique theoretical and algorithmic challenges [e.g., see 2, 7, 28]. It is therefore a natural approach for predicting scene graphs from images.

Structured prediction models typically define a score function $s(x, y)$ that quantifies how well a label assignment $y$ is compatible with an input $x$. In the case of understanding complex visual scenes, $x$ is an image, and $y$ is a complex label containing the labels of objects detected in an image and the labels of their relations. In this setup, the *inference task* amounts to finding the label that maximizes the compatibility score $y^* = \arg\max_y s(x, y)$. This score-based approach separates a scoring component – implemented by a parametric model, from an optimization component – aimed at finding a label that maximizes that score. Unfortunately, for a general scoring function $s(\cdot)$, the space of possible label assignments grows exponentially with input size. For instance, for scene graphs the set of possible object label assignments is too large even for relatively simple images, since the vocabulary of candidate objects may contain thousands of objects. As a result, inferring the label assignment that maximizes a scoring function is computationally hard in the general case.

An alternative approach to score-based methods is to map an input $x$ to a structured output $y$ with a "black box" neural network, without explicitly defining a score function. This raises a natural question: what is the right architecture for such a network? Here we take an axiomatic approach and argue that one important property such networks should satisfy is invariance to a particular type of input permutation. We then prove that this invariance is equivalent to imposing certain structural constraints on the architecture of the network, and describe architectures that satisfy these constraints.

To evaluate our approach, we first demonstrate on a synthetic dataset that respecting permutation invariance is important, because models that violate this invariance need more training data, despite having a comparable model size. Then, we tackle the problem of scene graph generation. We describe a model that satisfies the permutation invariance property, and show that it achieves state-of-the-art results on the competitive Visual Genome benchmark [15], demonstrating the power of our new design principle.

In summary, the novel contributions of this paper are: a) Deriving sufficient and necessary conditions for graph-permutation invariance in deep structured prediction architectures. b) Empirically demonstrating the benefit of graph-permutation invariance. c) Developing a state-of-the-art model for scene graph prediction on a large dataset of complex visual scenes.

## 2   Structured Prediction

Scored-based methods in structured prediction define a function $s(x, y)$ that quantifies the degree to which $y$ is compatible with $x$, and infer a label by maximizing $s(x, y)$ [e.g., see 2, 7, 16, 20, 28]. Most score functions previously used decompose as a sum over *simpler* functions, $s(x, y) = \sum_i f_i(x, y)$, making it possible to optimize $\max_y f_i(x, y)$ efficiently. This local maximization forms the basic building block of algorithms for approximately maximizing $s(x, y)$. One way to decompose the score function is to restrict each $f_i(x, y)$ to depend only on a small subset of the $y$ variables.

The renewed interest in deep learning led to efforts to integrate deep networks with structured prediction, including modeling the $f_i$ functions as deep networks. In this context, the most widely-used score functions are singleton $f_i(y_i, x)$ and pairwise $f_{ij}(y_i, y_j, x)$. The early work taking this approach used a two-stage architecture, learning the local scores independently of the structured prediction goal [6, 8]. Later studies considered *end-to-end* architectures where the inference algorithm

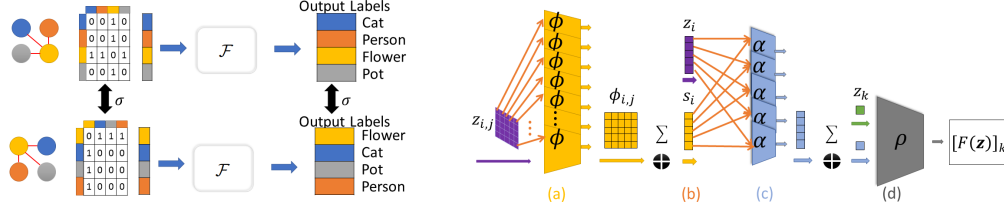

Figure 2: **Left**: Graph permutation invariance. A graph labeling function $\mathcal{F}$ is graph permutation invariant (GPI) if permuting the node features maintains the output. **Right**: a schematic representation of the GPI architecture in Theorem 1. Singleton features $z_i$ are omitted for simplicity. **(a)** First, the features $z_{i,j}$ are processed element-wise by $\phi$. **(b)** Features are summed to create a vector $s_i$, which is concatenated with $z_i$. **(c)** A representation of the entire graph is created by applying $\alpha$ $n$ times and summing the created vector. **(d)** The graph representation is then finally processed by $\rho$ together with $z_k$.

is part of the computation graph [7, 23, 26, 33]. Recent studies go beyond pairwise scores, also modelling global factors [2, 10].

Score-based methods provide several advantages. First, they allow intuitive specification of local dependencies between labels and how these translate to global dependencies. Second, for linear score functions, the learning problem has natural convex surrogates [16, 28]. Third, inference in large label spaces is sometimes possible via exact algorithms or empirically accurate approximations. However, with the advent of deep scoring functions $s(x, y; w)$, learning is no longer convex. Thus, it is worthwhile to rethink the architecture of structured prediction models, and consider models that map inputs $x$ to outputs $y$ directly without explicitly maximizing a score function. We would like these models to enjoy the expressivity and predictive power of neural networks, while maintaining the ability to specify local dependencies between labels in a flexible manner. In the next section, we present such an approach and consider a natural question: what should be the properties of a deep neural network used for structured prediction.

## 3  Permutation-Invariant Structured Prediction

In what follows we define the *permutation-invariance* property for structured prediction models, and argue that permutation invariance is a natural principle for designing their architecture.

We first introduce our notation. We focus on structures with pairwise interactions, because they are simpler in terms of notation and are sufficient for describing the structure in many problems. We denote a structured label by $y = [y_1, \ldots, y_n]$. In a score-based approach, the score is defined via a set of singleton scores $f_i(y_i, x)$ and pairwise scores $f_{ij}(y_i, y_j, x)$, where the overall score $s(x, y)$ is the sum of these scores. For brevity, we denote $f_{ij} = f_{ij}(y_i, y_j, x)$ and $f_i = f_i(y_i, x)$. An inference algorithm takes as input the local scores $f_i$, $f_{ij}$ and outputs an assignment that maximizes $s(x, y)$. We can thus view inference as a black-box that takes node-dependent and edge-dependent inputs (i.e., the scores $f_i$, $f_{ij}$) and returns a label $y$, even without an explicit score function $s(x, y)$. While numerous inference algorithms exist for this setup, including belief propagation (BP) and mean field, here we develop a framework for a deep labeling algorithm (we avoid the term "inference" since the algorithm does not explicitly maximize a score function). Such an algorithm will be a black-box, taking the $f$ functions as input and the labels $y_1, \ldots, y_n$ as output. We next ask what architecture such an algorithm should have.

We follow with several definitions. A *graph labeling function* $\mathcal{F} : (V, E) \to Y$ is a function whose input is an ordered set of node features $V = [z_1, \ldots, z_n]$ and an ordered set of edge features $E = [z_{1,2} \ldots, z_{i,j}, \ldots, z_{n,n-1}]$. For example, $z_i$ can be the array of values $f_i$, and $z_{i,j}$ can be the table of values $f_{i,j}$. Assume $z_i \in \mathbb{R}^d$ and $z_{i,j} \in \mathbb{R}^e$. The output of $\mathcal{F}$ is a set of node labels $y = [y_1, \ldots, y_n]$. Thus, algorithms such as BP are graph labeling functions. However, graph labeling functions do not necessarily maximize a score function. We denote the joint set of node features and edge features by $z$ (i.e., a set of $n + n(n-1) = n^2$ vectors). In Section 3.1 we discuss extensions to this case where only a subset of the edges is available.

A natural requirement is that the function $\mathcal{F}$ produces the same result when given the same features, up to a permutation of the input. For example, consider a label space with three variables $y_1, y_2, y_3$, and assume that $\mathcal{F}$ takes as input $z = (z_1, z_2, z_3, z_{12}, z_{13}, z_{23}) = (f_1, f_2, f_3, f_{12}, f_{13}, f_{23})$, and outputs a label $y = (y_1^*, y_2^*, y_3^*)$. When $\mathcal{F}$ is given an input that is permuted in a consistent way, say, $z' = (f_2, f_1, f_3, f_{21}, f_{23}, f_{13})$, this defines *exactly* the same input. Hence, the output should still be $y = (y_2^*, y_1^*, y_3^*)$. Most inference algorithms, including BP and mean field, satisfy this symmetry requirement by design, but this property is not guaranteed in general in a deep model. Here, our goal is to design a deep learning black-box, and hence we wish to guarantee invariance to input permutations. A black-box that violates this invariance "wastes" capacity on learning it at training time, which increases sample complexity, as shown in Sec. 5.1. We proceed to formally define the permutation invariance property.

**Definition 1.** *Let $z$ be a set of node features and edge features, and let $\sigma$ be a permutation of $\{1, \ldots, n\}$. We define $\sigma(z)$ to be a new set of node and edge features given by $[\sigma(z)]_i = z_{\sigma(i)}$ and $[\sigma(z)]_{i,j} = z_{\sigma(i),\sigma(j)}$.*

We also use the notation $\sigma([y_1, \ldots, y_n]) = [y_{\sigma(1)}, \ldots, y_{\sigma(n)}]$ for permuting the labels. Namely, $\sigma$ applied to a set of labels yields the same labels, only permuted by $\sigma$. Be aware that applying $\sigma$ to the input features is different from permuting labels, because edge input features must permuted in a way that is consistent with permuting node input features. We now provide our key definition of a function whose output is invariant to permutations of the input. See Figure 2 (left).

**Definition 2.** *A graph labeling function $\mathcal{F}$ is said to be **graph-permutation invariant** (GPI), if for all permutations $\sigma$ of $\{1, \ldots, n\}$ and for all $z$ it satisfies: $\mathcal{F}(\sigma(z)) = \sigma(\mathcal{F}(z))$.*

### 3.1 Characterizing Permutation Invariance

Motivated by the above discussion, we ask: what structure is necessary and sufficient to guarantee that $\mathcal{F}$ is GPI? Note that a function $\mathcal{F}$ takes as input an **ordered** set $z$. Therefore its output on $z$ could certainly differ from its output on $\sigma(z)$. To achieve permutation invariance, $\mathcal{F}$ should contain certain symmetries. For instance, one permutation invariant architecture could be to define $y_i = g(z_i)$ for any function $g$, but this architecture is too restrictive and does not cover all permutation invariant functions. Theorem 1 below provides a complete characterization (see Figure 2 for the corresponding architecture). Intuitively, the architecture in Theorem 1 is such that it can aggregate information from the entire graph, and do so in a permutation invariant manner.

**Theorem 1.** *Let $\mathcal{F}$ be a graph labeling function. Then $\mathcal{F}$ is graph-permutation invariant if and only if there exist functions $\alpha, \rho, \phi$ such that for all $k = 1, \ldots, n$:*

$$[\mathcal{F}(z)]_k = \rho \left( z_k, \sum_{i=1}^n \alpha \left( z_i, \sum_{j \neq i} \phi(z_i, z_{i,j}, z_j) \right) \right), \tag{1}$$

*where $\phi : \mathbb{R}^{2d+e} \to \mathbb{R}^L$, $\alpha : \mathbb{R}^{d+L} \to \mathbb{R}^W$ and $\rho : \mathbb{R}^{W+d} \to \mathbb{R}$.*

*Proof.* First, we show that any $\mathcal{F}$ satisfying the conditions of Theorem 1 is GPI. Namely, for any permutation $\sigma$, $[\mathcal{F}(\sigma(z))]_k = [\mathcal{F}(z)]_{\sigma(k)}$. To see this, write $[\mathcal{F}(\sigma(z))]_k$ using Eq. 1 and Definition 1:

$$[\mathcal{F}(\sigma(z))]_k = \rho(z_{\sigma(k)}, \sum_i \alpha(z_{\sigma(i)}, \sum_{j \neq i} \phi(z_{\sigma(i)}, z_{\sigma(i),\sigma(j)}, z_{\sigma(j)}))). \tag{2}$$

The second argument of $\rho$ above is invariant under $\sigma$, because it is a sum over nodes and their neighbors, which is invariant under permutation. Thus Eq. 2 is equal to:

$$\rho(z_{\sigma(k)}, \sum_i \alpha(z_i, \sum_{j \neq i} \phi(z_i, z_{i,j}, z_j))) = [\mathcal{F}(z)]_{\sigma(k)}$$

where equality follows from Eq. 1. We thus proved that Eq. 1 implies graph permutation invariance.

Next, we prove that *any* given GPI function $\mathcal{F}_0$ can be expressed as a function $\mathcal{F}$ in Eq. 1. Namely, we show how to define $\phi, \alpha$ and $\rho$ that can implement $\mathcal{F}_0$. Note that in this direction of the proof the function $\mathcal{F}_0$ is a black-box. Namely, we only know that it is GPI, but do not assume anything else about its implementation.

The key idea is to construct $\phi, \alpha$ such that the second argument of $\rho$ in Eq. 1 contains the information about *all* the graph features $z$. Then, the function $\rho$ corresponds to an application of $\mathcal{F}_0$ to this representation, followed by extracting the label $y_k$. To simplify notation assume edge features are scalar ($e = 1$). The extension to vectors is simple, but involves more indexing.

We assume WLOG that the black-box function $\mathcal{F}_0$ is a function only of the pairwise features $z_{i,j}$ (otherwise, we can always augment the pairwise features with the singleton features). Since $z_{i,j} \in \mathbb{R}$ we use a matrix $\mathbb{R}^{n,n}$ to denote all the pairwise features.

Finally, we assume that our implementation of $\mathcal{F}_0$ will take additional node features $z_k$ such that no two nodes have the same feature (i.e., the features identify the node).

Our goal is thus to show that there exist functions $\alpha, \phi, \rho$ such that the function in Eq. 2 applied to $Z$ yields the same labels as $\mathcal{F}_0(Z)$.

Let $H$ be a hash function with $L$ buckets mapping node features $z_i$ to an index (bucket). Assume that $H$ is perfect (this can be achieved for a large enough $L$). Define $\phi$ to map the pairwise features to a vector of size $L$. Let $\mathbb{1}[j]$ be a one-hot vector of dimension $\mathbb{R}^L$, with one in the $j^{\text{th}}$ coordinate. Recall that we consider scalar $z_{i,j}$ so that $\phi$ is indeed in $\mathbb{R}^L$, and define $\phi$ as: $\phi(z_i, z_{i,j}, z_j) = \mathbb{1}[H(z_j)] z_{i,j}$, i.e., $\phi$ "stores" $z_{i,j}$ in the unique bucket for node $j$.

Let $s_i = \sum_{z_{i,j} \in E} \phi(z_i, z_{i,j}, z_j)$ be the second argument of $\alpha$ in Eq. 1 ($s_i \in \mathbb{R}^L$). Then, since all $z_j$ are distinct, $s_i$ stores all the pairwise features for neighbors of $i$ in unique positions within its $L$ coordinates. Since $s_i(H(z_k))$ contains the feature $z_{i,k}$ whereas $s_j(H(z_k))$ contains the feature $z_{j,k}$, we cannot simply sum the $s_i$, since we would lose the information of which edges the features originated from. Instead, we define $\alpha$ to map $s_i$ to $\mathbb{R}^{L \times L}$ such that each feature is mapped to a distinct location. Formally:

$$\alpha(z_i, s_i) = \mathbb{1}[H(z_i)] s_i^T . \tag{3}$$

$\alpha$ outputs a matrix that is all zeros except for the features corresponding to node $i$ that are stored in row $H(z_i)$. The matrix $M = \sum_i \alpha(z_i, s_i)$ (namely, the second argument of $\rho$ in Eq. 1) is a matrix with all the edge features in the graph including the graph structure.

To complete the construction we set $\rho$ to have the same outcome as $\mathcal{F}_0$. We first discard rows and columns in $M$ that do not correspond to original nodes (reducing $M$ to dimension $n \times n$). Then, we use the reduced matrix as the input $z$ to the black-box $\mathcal{F}_0$.

Assume for simplicity that $M$ does not need to be contracted (this merely introduces another indexing step). Then $M$ corresponds to the original matrix $Z$ of pairwise features, with both rows and columns permuted according to $H$. We will thus use $M$ as input to the function $\mathcal{F}_0$. Since $\mathcal{F}_0$ is GPI, this means that the label for node $k$ will be given by $\mathcal{F}_0(M)$ in position $H(z_k)$. Thus we set $\rho(z_k, M) = [\mathcal{F}_0(M)]_{H(z_k)}$, and by the argument above this equals $[\mathcal{F}_0(Z)]_k$, implying that the above $\alpha, \phi$ and $\rho$ indeed implement $\mathcal{F}_0$. $\square$

**Extension to general graphs**  So far, we discussed complete graphs, where edges correspond to valid feature pairs. However, many graphs of interest might be incomplete. For example, an $n$-variable chain graph in sequence labeling has only $n - 1$ edges. For such graphs, the input to $\mathcal{F}$ would not contain all $z_{i,j}$ pairs but rather only features corresponding to valid edges of the graph, and we are only interested in invariances that preserve the graph structure, namely, the automorphisms of the graph. Thus, the desired invariance is that $\sigma(\mathcal{F}(z)) = \mathcal{F}(\sigma(z))$, where $\sigma$ is not an arbitrary permutation but an automorphism. It is easy to see that a simple variant of Theorem 1 holds in this case. All we need to do is replace in Eq. 2 the sum $\sum_{j \neq i}$ with $\sum_{j \in N(i)}$, where $N(i)$ are the neighbors of node i in the graph. The arguments are then similar to the proof above.

**Implications of Theorem 1**  Our result has interesting implications for deep structured prediction. First, it highlights that the fact that the architecture "collects" information from *all* different edges of the graph, in an invariant fashion via the $\alpha, \phi$ functions. Specifically, the functions $\phi$ (after summation) aggregate all the features around a given node, and then $\alpha$ (after summation) can collect them. Thus, these functions can provide a summary of the entire graph that is sufficient for downstream algorithms. This is different from one round of message passing algorithms which would not be sufficient for collecting global graph information. Note that the dimensions of $\phi, \alpha$ may need to be large to aggregate all graph information (e.g., by hashing all the features as in the proof of Theorem 1), but the architecture itself can be shallow.

Second, the architecture is parallelizable, as all $\phi$ functions can be applied simultaneously. This is in contrast to recurrent models [32] which are harder to parallelize and are thus slower in practice.

Finally, the theorem suggests several common architectural structures that can be used within GPI. We briefly mention two of these. 1) **Attention:** Attention is a powerful component in deep learning architectures [1], but most inference algorithms do not use attention. Intuitively, in attention each node $i$ aggregates features of neighbors through a weighted sum, where the weight is a function of the neighbor's relevance. For example, the label of an entity in an image may depend more strongly on entities that are spatially closer. Attention can be naturally implemented in our GPI characterization, and we provide a full derivation for this implementation in the appendix. It plays a key role in our scene graph model described below. 2) **RNNs:** Because GPI functions are closed under composition, for any GPI function $\mathcal{F}$ we can run $\mathcal{F}$ iteratively by providing the output of one step of $\mathcal{F}$ as part of the input to the next step and maintain GPI. This results in a recurrent architecture, which we use in our scene graph model.

## 4   Related Work

The concept of architectural invariance was recently proposed in DEEPSETS [31]. The invariance we consider is much less restrictive: the architecture does not need to be invariant to all permutations of singleton and pairwise features, just those consistent with a graph re-labeling. This characterization results in a substantially different set of possible architectures.

**Deep structured prediction**. There has been significant recent interest in extending deep learning to structured prediction tasks. Much of this work has been on semantic segmentation, where convolutional networks [27] became a standard approach for obtaining "singleton scores" and various approaches were proposed for adding structure on top. Most of these approaches used variants of message passing algorithms, unrolled into a computation graph [29]. Some studies parameterized parts of the message passing algorithm and learned its parameters [18]. Recently, gradient descent has also been used for maximizing score functions [2, 10]. An alternative to deep structured prediction is greedy decoding, inferring each label at a time based on previous labels. This approach has been popular in sequence-based applications (e.g., parsing [5]), relying on the sequential structure of the input, where BiLSTMs are effectively applied. Another related line of work is applying deep learning to graph-based problems, such as TSP [3, 9, 13]. Clearly, the notion of graph invariance is important in these, as highlighted in [9]. They however do not specify a general architecture that satisfies invariance as we do here, and in fact focus on message passing architectures, which we strictly generalize. Furthermore, our focus is on the more general problem of structured prediction, rather than specific graph-based optimization problems.

**Scene graph prediction.** Extracting scene graphs from images provides a semantic representation that can later be used for reasoning, question answering, and image retrieval [12, 19, 25]. It is at the forefront of machine vision research, integrating challenges like object detection, action recognition and detection of human-object interactions [17, 24]. Prior work on scene graph predictions used neural message passing algorithms [29] as well as prior knowledge in the form of word embeddings [19]. Other work suggested to predict graphs directly from pixels in an end-to-end manner [21]. NeuralMotif [32], currently the state-of-the-art model for scene graph prediction on Visual Genome, employs an RNN that provides global context by sequentially reading the independent predictions for each entity and relation and then refines those predictions. The NEURALMOTIF model maintains GPI by fixing the order in which the RNN reads its inputs and thus only a single order is allowed. However, this fixed order is not guaranteed to be optimal.

## 5   Experimental Evaluation

We empirically evaluate the benefit of GPI architectures. First, using a synthetic graph-labeling task, and then for the problem of mapping images to scene graphs.

### 5.1   Synthetic Graph Labeling

We start with studying GPI on a synthetic problem, defined as follows. An input graph $G = (V, E)$ is given, where each node $i \in V$ is assigned to one of $K$ sets. The set for node $i$ is denoted by

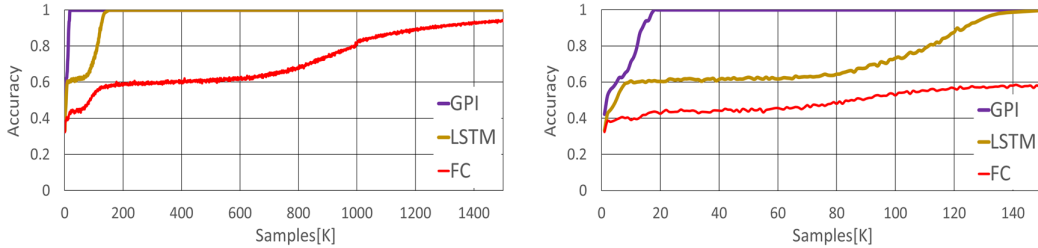

Figure 3: Accuracy as a function of sample size for graph labeling. Right is a zoomed in version of left.

$\Gamma(i)$. The goal is to compute for each node the number of neighbors that belong to the same set. Namely, the label of a node is $y_i = \sum_{j \in N(i)} \mathbb{1}[\Gamma(i) = \Gamma(j)]$. We generated random graphs with 10 nodes (larger graphs produced similar results) by sampling each edge independently and uniformly, and sampling $\Gamma(i)$ for every node uniformly from $\{1, \ldots, K\}$. The node features $z_i \in \{0, 1\}^K$ are one-hot vectors of $\Gamma(i)$ and the edge features $z_{i,j} \in \{0, 1\}$ indicate whether $ij \in E$. We compare two standard non-GPI architectures and one GPI architecture: (a) A GPI-architecture for graph prediction, described in detail in Section 5.2. We used the basic version without attention and RNN. (b) LSTM: We replace $\sum \phi(\cdot)$ and $\sum \alpha(\cdot)$, which perform aggregation in Theorem 1 with two LSTMs with a state size of 200 that read their input in random order. (c) A fully-connected (FC) feed-forward network with 2 hidden layers of 1000 nodes each. The input to the fully connected model is a concatenation of all node and pairwise features. The output is all node predictions. The focus of the experiment is to study sample complexity. Therefore, for a fair comparison, we use the same number of parameters for all models.

Figure 3, shows the results, demonstrating that GPI requires far fewer samples to converge to the correct solution. This illustrates the advantage of an architecture with the correct inductive bias for the problem.

## 5.2 Scene-Graph Classification

We evaluate the GPI approach on the motivating task of this paper, inferring scene graphs from images (Figure 1). In this problem, the input is an image annotated with a set of *bounding boxes* for the entities in the image.[2] The goal is to label each bounding box with the correct entity category and every pair of entities with their relation, such that they form a coherent *scene graph*.

We begin by describing our *Scene Graph Predictor (SGP)* model. We aim to predict two types of variables. The first is entity variables $[y_1, \ldots, y_n]$ for all bounding boxes. Each $y_i$ can take one of $L$ values (e.g., "dog", "man"). The second is relation variables $[y_{n+1}, \ldots, y_{n^2}]$ for every pair of bounding boxes. Each such $y_j$ can take one of $R$ values (e.g., "on", "near"). Our graph connects variables that are expected to be inter-related. It contains two types of edges: 1) **entity-entity edge** connecting every two entity variables ($y_i$ and $y_j$ for $1 \leq i \neq j \leq n$. 2) **entity-relation edges** connecting every relation variable $y_k$ (where $k > n$) to its two entity variables. Thus, our graph is not a complete graph and our goal is to design an architecture that will be invariant to any automorphism of the graph, such as permutations of the entity variables.

For the input features $z$, we used the features learned by the baseline model from [32].[3] Specifically, the entity features $z_i$ included (1) The confidence probabilities of all entities for $y_i$ as learned by the baseline model. (2) Bounding box information given as (`left, bottom, width, height`); (3) The number of smaller entities (also bigger); (4) The number of entities to the left, right, above and below. (5) The number of entities with higher and with lower confidence; (6) For the linguistic model only: word embedding of the most probable class. Word vectors were learned with GLOVE from the ground-truth captions of Visual Genome.

Similarly, the relation features $z_j \in \mathbb{R}^R$ contained the probabilities of relation entities for the relation $j$. For the Linguistic model, these features were extended to include word embedding of the most probable class. For entity-entity pairwise features $z_{i,j}$, we use the relation probability for each pair.

| | Constrained Evaluation | | | | Unconstrained Evaluation | | | |
| --- | --- | --- | --- | --- | --- | --- | --- | --- |
| | SGCls | | PredCls | | SGCls | | PredCls | |
| | R@50 | R@100 | R@50 | R@100 | R@50 | R@100 | R@50 | R@100 |
| Lu et al., 2016 [19] | 11.8 | 14.1 | 35.0 | 27.9 | - | - | - | - |
| Xu et al., 2017 [29] | 21.7 | 24.4 | 44.8 | 53.0 | - | - | - | - |
| Pixel2Graph [21] | - | - | - | - | 26.5 | 30.0 | 68.0 | 75.2 |
| Graph R-CNN [30] | 29.6 | 31.6 | 54.2 | 59.1 | - | - | - | - |
| Neural Motifs [32] | 35.8 | 36.5 | **65.2** | **67.1** | 44.5 | 47.7 | **81.1** | **88.3** |
| Baseline [32] | 34.6 | 35.3 | 63.7 | 65.6 | 43.4 | 46.6 | 78.8 | 85.9 |
| No Attention | 35.3 | 37.2 | 64.5 | 66.3 | 44.1 | 48.5 | 79.7 | 86.7 |
| Neighbor Attention | 35.7 | 38.5 | 64.6 | 66.6 | 44.7 | 49.9 | 80.0 | 87.1 |
| Linguistic | **36.5** | **38.8** | 65.1 | 66.9 | **45.5** | **50.8** | 80.8 | 88.2 |

Table 1: Test set results for graph-constrained evaluation (i.e., the returned triplets must be consistent with a scene graph) and for unconstrained evaluation (triplets need not be consistent with a scene graph).

Because the output of SGP are probability distributions over entities and relations, we use them as an the input $z$ to SGP, once again in a recurrent manner and maintain GPI.

We next describe the main components of the GPI architecture. First, we focus on the parts that output the entity labels. $\phi_{ent}$ is the network that integrates features for two entity variables $y_i$ and $y_j$. It simply takes $z_i$, $z_j$ and $z_{i,j}$ as input, and outputs a vector of dimension $n_1$. Next, the network $\alpha_{ent}$ takes as input the outputs of $\phi_{ent}$ for all neighbors of an entity, and uses the attention mechanism described above to output a vector of dimension $n_2$. Finally, the $\rho_{ent}$ network takes these $n_2$ dimensional vectors and outputs $L$ logits predicting the entity value. The $\rho_{rel}$ network takes as input the $\alpha_{ent}$ representation of the two entities, as well as $z_{i,j}$ and transforms the output into $R$ logits. See appendix for specific network architectures.

### 5.2.1 Experimental Setup and Results

**Dataset.** We evaluated our approach on Visual Genome (VG) [15], a dataset with 108,077 images annotated with bounding boxes, entities and relations. On average, images have 12 entities and 7 relations per image. For a proper comparison with previous results [21, 29, 32], we used the data from [29], including the train and test splits. For evaluation, we used the same 150 entities and 50 relations as in [21, 29, 32]. To tune hyper-parameters, we also split the training data into two by randomly selecting 5K examples, resulting in a final 70K/5K/32K split for train/validation/test sets.

**Training.** All networks were trained using Adam [14] with batch size 20. Hyperparameter values below were chosen based on the validation set. The SGP loss function was the sum of cross-entropy losses over all entities and relations in the image. In the loss, we penalized entities 4 times more strongly than relations, and penalized negative relations 10 times more weakly than positive relations.

**Evaluation.** In [29] three different evaluation settings were considered. Here we focus on two of these: **(1) SGCls:** Given ground-truth bounding boxes for entities, predict all entity categories and relations categories. **(2) PredCls:** Given bounding boxes annotated with entity labels, predict all relations. Following [19], we used Recall@$K$ as the evaluation metric. It measures the fraction of correct ground-truth triplets that appear within the $K$ most confident triplets proposed by the model. Two evaluation protocols are used in the literature differing in whether they enforce graph constraints over model predictions. The first *graph-constrained* protocol requires that the top-$K$ triplets assign one consistent class per entity and relation. The second *unconstrained* protocol does not enforce any such constraints. We report results on both protocols, following [32].

**Models and baselines.** We compare four variants of our GPI approach with the reported results of four baselines that are currently the state-of-the-art on various scene graph prediction problems (all models use the same data split and pre-processing as [29]): 1) LU ET AL., 2016 [19]: This work leverages word embeddings to fine-tune the likelihood of predicted relations. 2) XU ET AL, 2017 [29]: This model passes messages between entities and relations, and iteratively refines the feature map used for prediction. 3) NEWELL & DENG, 2017 [21]: The PIXEL2GRAPH model uses associative embeddings [22] to produce a full graph from the image. 4) YANG ET AL., 2018 [30]: The GRAPH R-CNN model uses object-relation regularities to sparsify and reason over scene graphs. 5) ZELLERS ET AL., 2017 [32]: The NEURALMOTIF method encodes global context for

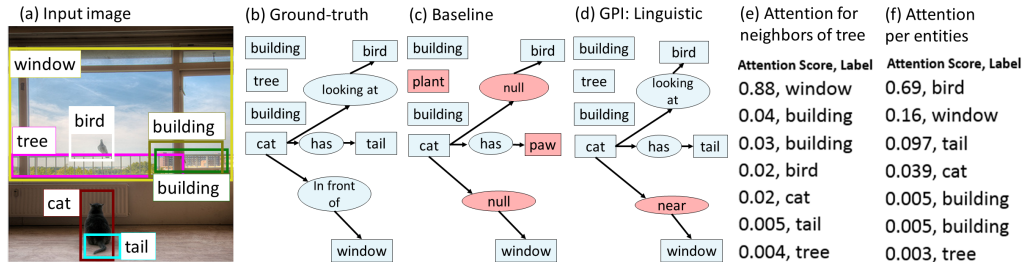

Figure 4: (a) An input image with bounding boxes from VG. (b) The ground-truth scene graph. (c) The Baseline fails to recognize some entities (*tail* and *tree*) and relations (*in front of* instead of *looking at*). (d) GPI:LINGUISTIC fixes most incorrect LP predictions. (e) *Window* is the most significant neighbor of *Tree*. (f) The entity *bird* receives substantial attention, while *Tree* and *building* are less informative.

capturing high-order motifs in scene graphs, and the BASELINE outputs the entities and relations distributions without using the global context. The following variants of GPI were compared: 1) GPI: NO ATTENTION: Our GPI model, but with no attention mechanism. Instead, following Theorem 1, we simply sum the features. 2) GPI: NEIGHBORATTENTION: Our GPI model, with attention over neighbors features. 3) GPI: LINGUISTIC: Same as GPI: NEIGHBORATTENTION but also concatenating the word embedding vector, as described above.

**Results.** Table 1 shows recall@50 and recall@100 for three variants of our approach, and compared with five baselines. All GPI variants performs well, with LINGUISTIC outperforming all baselines for SGCls and being comparable to the state-of-the-art model for PredCls. Note that PredCl is an easier task, which makes less use of the structure, hence it is not surprising that GPI achieves similar accuracy to [32]. Figure 4 illustrates the model behavior. Predicting isolated labels with $z_i$ (4c) mislabels several entities, but these are corrected at the final output (4d). Figure 4e shows that the system learned to attend more to nearby entities (the window and building are closer to the tree), and 4f shows that stronger attention is learned for the class bird, presumably because it is usually more informative than common classes like tree.

**Implementation details.** The $\phi$ and $\alpha$ networks were each implemented as a single fully-connected (FC) layer with a 500-dimensional outputs. $\rho$ was implemented as a FC network with 3 500-dimensional hidden layers, with one 150-dimensional output for the entity probabilities, and one 51-dimensional output for relation probabilities. The attention mechanism was implemented as a network like to $\phi$ and $\alpha$, receiving the same inputs, but using the output scores for the attention . The full code is available at https://github.com/shikorab/SceneGraph

# 6 Conclusion

We presented a deep learning approach to structured prediction, which constrains the architecture to be invariant to structurally identical inputs. As in score-based methods, our approach relies on pairwise features, capable of describing inter-label correlations, and thus inheriting the intuitive aspect of score-based approaches. However, instead of maximizing a score function (which leads to computationally-hard inference), we directly produce an output that is invariant to equivalent representations of the pairwise terms.

This axiomatic approach to model architecture can be extended in many ways. For image labeling, geometric invariances (shift or rotation) may be desired. In other cases, invariance to feature permutations may be desirable. We leave the derivation of the corresponding architectures to future work. Finally, there may be cases where the invariant structure is unknown and should be discovered from data, which is related to work on lifting graphical models [4]. It would be interesting to explore algorithms that discover and use such symmetries for deep structured prediction.

## Acknowledgements

This work was supported by the ISF Centers of Excellence grant, and by the Yandex Initiative in Machine Learning. Work by GC was performed while at Google Brain Research.

## Footnotes

[2]For simplicity, we focus on the task where boxes are given.

[3]The baseline does not use any LSTM or context, and is thus unrelated to the main contribution of [32].

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
