[Supplementary Material]

# Supplemental Material - Mapping Images to Scene Graphs with Permutation-Invariant Structured Prediction

**Roei Herzig**[*]
Tel Aviv University
roeiherzig@mail.tau.ac.il

**Moshiko Raboh**[*]
Tel Aviv University
mosheraboh@mail.tau.ac.il

**Gal Chechik**
Bar-Ilan University, NVIDIA Research
gal.chechik@biu.ac.il

**Jonathan Berant**
Tel Aviv University, AI2
joberant@cs.tau.ac.il

**Amir Globerson**
Tel Aviv University
gamir@post.tau.ac.il

This supplementary material includes: (1) Visual illustration of the proof of Theorem 1. (2) Explaining how to integrate an attention mechanism in our GPI framework. (3) Additional evaluation method to further analyze and compare our work with baselines.

## 1 Theorem 1: Illustration of Proof

Figure 1: Illustration of the proof of Theorem 1 using a specific construction example. Here $H$ is a hash function of size $L = 5$ such that $H(1) = 1, H(3) = 2, H(2) = 4$, $G$ is a three-node input graph, and $\boldsymbol{z}_{i,j} \in \mathbb{R}$ are the pairwise features (in purple) of $G$. **(a)** $\phi$ is applied to each $\boldsymbol{z}_{i,j}$. Each application yields a vector in $\mathbb{R}^5$. The three dark yellow columns correspond to $\phi(\boldsymbol{z}_{1,1})$, $\phi(\boldsymbol{z}_{1,2})$ and $\phi(\boldsymbol{z}_{1,3})$. Then, all vectors $\phi(\boldsymbol{z}_{i,j})$ are summed over $j$ to obtain three $\boldsymbol{s}_i$ vectors. **(b)** $\boldsymbol{\alpha}$'s (blue matrices) are an outer product between $\mathbb{1}\left[H(\boldsymbol{z}_i)\right]$ and $\boldsymbol{s}_i$ resulting in a matrix of zeros except one row. The dark blue matrix corresponds for $\alpha(\boldsymbol{z}_1, \boldsymbol{s}_1)$. **(c)** All $\alpha$'s are summed to a $5 \times 5$ matrix, isomorphic to the original $\boldsymbol{z}_{i,j}$ matrix.

---

[*]Equal Contribution.

## 2 Characterizing Permutation Invariance: Attention

Attention is a powerful component which naturally can be introduced into our GPI model. We now show how attention can be introduced in our framework. Formally, we learn attention weights for the neighbors $j$ of a node $i$, which scale the features $z_{i,j}$ of that neighbor. We can also learn different attention weights for individual features of each neighbor in a similar way.

Let $w_{i,j} \in \mathbb{R}$ be an attention mask specifying the weight that node $i$ gives to node $j$:

$$w_{i,j}(z_i, z_{i,j}, z_j) = e^{\beta(z_i, z_{i,j}, z_j)} / \sum_t e^{\beta(z_i, z_{i,t}, z_t)} \tag{1}$$

where $\beta$ can be any scalar-valued function of its arguments (e.g., a dot product of $z_i$ and $z_j$ as in standard attention models). To introduce attention we wish $\alpha \in \mathbb{R}^e$ to have the form of weighting $w_{i,j}$ over neighboring feature vectors $z_{i,j}$, namely, $\alpha = \sum_{j \neq i} w_{i,j} z_{i,j}$.

To achieve this form we extend $\phi$ by a single entry, defining $\phi \in \mathbb{R}^{e+1}$ (namely we set $L = e + 1$) as $\phi_{1:e}(z_i, z_{i,j}, z_j) = e^{\beta(z_i, z_{i,j}, z_j)} z_{i,j}$ (here $\phi_{1:e}$ are the first $e$ elements of $\phi$) and $\phi_{e+1}(z_i, z_{i,j}; z_j) = e^{\beta(z_i, z_{i,j}, z_j)}$. We keep the definition of $s_i = \sum_{j \neq i} \phi(z_i, z_{i,j}, z_j)$. Next, we define $\alpha = \frac{s_{i,1:e}}{s_{i,e+1}}$ and substitute $s_i$ and $\phi$ to obtain the desired form as attention weights $w_{i,j}$ over neighboring feature vectors $z_{i,j}$:

$$\alpha(z_i, s_i) = \frac{s_{i,1:e}}{s_{i,e+1}} = \sum_{j \neq i} \frac{e^{\beta(z_i, z_{i,j}, z_j)} z_{i,j}}{\sum_{j \neq i} e^{\beta(z_i, z_{i,j}, z_j)}} = \sum_{j \neq i} w_{i,j} z_{i,j}$$

A similar approach can be applied over $\alpha$ and $\rho$ to model attention over the outputs of $\alpha$ as well (graph nodes).

## 3 Scene Graph Results

In the main paper, we described the results for the two prediction tasks: SGCls and PredCls, as defined in section 5.2.1: "Experimental Setup and Results". To further analyze our module, we compare the best variant, GPI: LINGUISTIC, per relation to two baselines: [1] and [2]. Table 1, specifies the PredCls recall@5 of the 20-top frequent relation classes. The GPI module performs better in almost all the relations classes.

Table 1: Recall@5 of PredCls for the 20-top relations ranked by their frequency, as in [2]

| RELATION | [1] | [2] | LINGUISTIC |
|---|---|---|---|
| ON | **99.71** | 99.25 | 99.3 |
| HAS | 98.03 | 97.25 | **98.7** |
| IN | 80.38 | 88.30 | **95.9** |
| OF | 82.47 | 96.75 | **98.1** |
| WEARING | 98.47 | 98.23 | **99.6** |
| NEAR | 85.16 | **96.81** | 95.4 |
| WITH | 31.85 | 88.10 | **94.2** |
| ABOVE | 49.19 | 79.73 | **83.9** |
| HOLDING | 61.50 | 80.67 | **95.5** |
| BEHIND | 79.35 | **92.32** | 91.2 |
| UNDER | 28.64 | 52.73 | **83.2** |
| SITTING ON | 31.74 | 50.17 | **90.4** |
| IN FRONT OF | 26.09 | 59.63 | **74.9** |
| ATTACHED TO | 8.45 | 29.58 | **77.4** |
| AT | 54.08 | 70.41 | **80.9** |
| HANGING FROM | 0.0 | 0.0 | **74.1** |
| OVER | 9.26 | 0.0 | **62.4** |
| FOR | 12.20 | 31.71 | **45.1** |
| RIDING | 72.43 | 89.72 | **96.1** |