[Reviews · NeurIPS 2018]

Reviewer 1



This paper studies the property of permutation invariance in the context of structured prediction. The paper argues that in many applications permutation invariance is a desirable property of a solution and it makes sense to design the model such that it is satisfied by construction rather than to rely on learning to get this property. The paper proposes a model to represent permutation invariant functions and claims that this model is a universal approximator within this family. The proposed method is evaluated on a synthetic and a real task (labelling of scene graphs). 1) Most importantly, I think that in the current form the proof of the main theoretical result (Theorem 1) is wrong. The problem is with the reverse direction proving that any permutation invariant function can be represented in the form of Theorem 1. Specifically, Lines 142-159 construct matrix M which aggregates information about the graph edges. However, the actual function \rho is defined (line 160) without taking the matrix M into account. Instead, it looks at y, which is the output of the function. As is, the statement in fact says "define the function output with y, now set it equals to y", which is a logical error and does not prove anything. 2) I also think that the assumptions that z_k uniquely identifies the node (line 138) and that F is a function of only pairwise features (line 140) contradict each other, because if there are no unary features they cannot identify the nodes. 3) The form of the function F defined in Theorem 1 is very similar to one round of neural message passing [9] and thus will not allow to propagate information across the graph. For example, imaging the toy situation: one node has a very strong preference for one label through the unary potentials, the others have no preferences. The pairwise potentials defined a long chain of equalities saying that in the end of the day all the variables should take the same value. However, to infer this we need to propagate information across the whole graph, which will need many rounds. 4) The experimental evaluation is somewhat interesting, but does not have sufficient description to be reproduced and does not show significant improvements, thus I find it not sufficient for acceptance. Minor: Line 147 probably should have the sum over j and not over pairs i, j ==== after the response 1) I agree with the authors my point 1 was not valid. 2) I agree with the authors that this particular place has no contradiction. However, I'm still not quite buying the whole argument based on the perfect hash function. The theorem is proving a statement about continuous inputs, which means that for any hash function with a finite hash table there exist a pair of inputs z_i, z_j, for which there will be a collision. The construction will break for this input. I think we cannot use a different hash function for each input, but need it to defined before looking at the inputs, which looks like a fundamental flaw in the proof technique. 3) I'm not getting why the presented method cannot be viewed as a message passing algorithm. Functions phi can be viewed as messages and alpha as the message aggregation. The big difference is probably the fact that the messages are not passed w.r.t. the initial graph, but w.r.t. the complete graph on the nodes. 4) I still think that the description of the experiments does not allow reproducibility and the improvements are somewhat incremental (the response claims otherwise, but does not give any extra data, e.g. some upper bounds). Having said that, I'm ready to raise my score to 5, because the theorem might be correct, but the paper still requires a major revision (proof of the theorem and the description of the experimental setup).

Reviewer 2



This paper focuses on designing neural architectures for problems which exhibit the structured output's invariance to permutations of the sub-structures scores and show empirical results on a synthetic task and scene graph labeling task. The proposed architecture is fairly simple and imposes minor restrictions on the kind of parametrization required to achieve this invariance. The empirical results suggest that their approach makes use of the inductive bias regarding permutation. A concern is that the technique proposed in the paper seems fairly limited to the problems where the desired invariance properties are known and easily expressible in terms of input nodes and factors. In many cases, the invariance is either not exactly known (and should be learned) or is hard to express simply in terms of the individual input factors. For example, even rotational invariance in images might be tricky to characterize with the proposed approach. Overall, the motivation behind the problem is clear and the approach is desirable for an important class of problems in which the output subparts are correlated but not necessarily ordered or sensitive to permutation (eg set prediction). Presentation: The proof for sufficiency (any GPI can be represented by eq1) was difficult to follow. Please make it clearer, possibly by using illustrations. Also, the note on line 185 about interaction of RNNs and the proposed approach seems out of place and I'm not sure if any experiments were done regarding the comment on this note.

Reviewer 3



Main idea: The paper tackles the problem of scene graph prediction -- a special problem in structured prediction. As in many deep learning models for structured prediction, the central problem is what network architecture should be used. This paper argues that the network should be permutation-invariant wrt inputs, and thus proposes an architecture satisfied such kind of property (+ attention and recurrent forward). Experiments show that the proposed method is comparable with the state-of-the-art. Strength: - The paper is well written. Most of the details of the model and training are covered. It will be great if the authors publish their code for reproducibility. - The experiments are well organized. Outperformance has shown in terms of the criterion SGCI. Weakness: - The motivation to introduce permutation invariance to structured prediction is not clear. I understand this is a particular inductive bias that the authors would like to impose, but how is this property related to structured prediction? in terms of either learning theory or empirical observations? Or it may be more interesting to know how do the authors come up with this idea. - Error in Theorem 1: The direction (1) => graph-permutation invariant (GPI) is easy to understand. However, I don't think GPI => (1) holds true in general (any GPI function can be expressed as in (1) sounds like a strong statement). The proof for this direction is really hard to follow. Even if this proof is correct, a major revision is needed. As a counter example, let's construct a graph-labeling function with each element of its output the same = \sum_i \sum_{j \neq i} \phi_ij(z_ij). This function is GPI by definition. E.g., let \phi_ij = \phi^{i*j}. Note that in (1), \phi is shared by all ij, and \alpha only takes \sum \phi(z_ij) as its 2nd argument, so we cannot introduce \phi_ij by \alpha. If I'm not mistaken, there is no way to express this counter example in the form of (1). After rebuttal: The rebuttal addressed some of my concerns, especially the counter example as pointed out doesn't hold true. I tend to believe that graph-permutation invariance could be a case of inductive bias for structured prediction, which deserves to be examined by more audiences. The major concern now is the writing. It still sounds a strong statement to me that any graph-permutation invariant function can be expressed as in (1). Rather than presenting the idea by a compact proof. A major rewriting for showing the intuition of (1) would be very helpful. For example, to show that message-passing algorithms can be considered as special cases.